# Physicochemical Properties and Effects of Fruit Pulps from the Amazon Biome on Physiological Parameters in Rats

**DOI:** 10.3390/nu13051484

**Published:** 2021-04-28

**Authors:** Fernanda Rosan Fortunato Seixas, Bruna Kempfer Bassoli, Lara Borghi Virgolin, Laís Chancare Garcia, Natália Soares Janzantti

**Affiliations:** 1Department of Health Science, Federal University of Grande Dourados, Highway Dourados/Itahum, Km 12—Unit II, 79804-970 Dourados, Brazil; laisgarcianutri@gmail.com; 2Department of Engineering and Food Technology, São Paulo State University, R. Cristóvão Colombo, 2265—Jardim Nazareth, 15054-000 São José do Rio Preto, Brazil; laraborghi.lb@gmail.com (L.B.V.); natalia.soares-janzantti@unesp.br (N.S.J.); 3Medical School, Federal University of Roraima, Avenida Capitão Ene Garcês, n° 2413—Aeroporto, 69310-000 Boa Vista, Brazil; bruna.bassoli@ufrr.br

**Keywords:** Amazonian fruits, composition, metabolic effects

## Abstract

This study aimed to analyze the physicochemical characteristics and the effects of Amazonian pulp fruits consumption, such as araçá-boi (*Eugenia stipitata*), abiu grande (*Pouteria caimito*), araticum (*Annona crassiflora*), biri-biri (*Averrhoa bilimbi* L.), and yellow mangosteen (*Garcinia xanthochymus*), on hematologic, metabolic, renal, and hepatic function parameters in Wistar rats (*n* = 10 rats/group). The pulp of abiu had the highest levels of soluble solids, sugars, and pH. Biri-biri pulp had the highest levels of ascorbic acid and total titratable acidity, and a low pH. The araticum pulp had higher (*p* ≤ 0.05) ash content, total phenolic compounds, and antioxidant activity than the pulp of other analyzed fruits. No significant increase in hematocrit, nor reduction of blood glucose, plasma cholesterol, and serum levels of glutamic-pyruvic transaminase (TGP), creatinine, and urea was observed in experimental groups relative to the control group of rats after the consumption of fruits pulp. The intake of abiu and araticum pulps promoted a significant reduction (*p* ≤ 0.05) in total leukocytes of the experimental groups as compared to the control group and only the intake of araticum significantly increased (*p* ≤ 0.05) triglyceride blood levels in rats (99.50 mg/dL). The regular consumption of biri-biri pulp for 30 days significantly (*p* ≤ 0.05) increased serum glutamic-oxaloacetic transaminase (TGO) levels in rats (116.83 U/L) compared to the control group (98.00 U/L). More researches are needed to generate knowledge about these promising Amazonian fruits, supporting the native fruit production, in addition to promoting health in the population and sustainability in the Amazon region.

## 1. Introduction

Brazil is the third largest producer of fruits in the world, and Amazonian fruits are known to contain bioactive compounds. However, despite their high nutritional value and potential for promoting health benefits, these fruits are still relatively unexplored [1,2,3]. The production and commercialization of açai is currently notable in the Amazon biome, due to its bioactive and antioxidant potential [4]. However, there are other fruits in this biome with similar economic and nutritional importance for the local population, and that have good marketing potential. These include araçá-boi (*Eugenia stipitata*), abiu grande (*Pouteria caimito*), araticum (*Annona crassiflora*), biri-biri (*Averrhoa bilimbi* L.), and yellow mangosteen (*Garcinia xanthochymus*), which can be consumed fresh or in the form of juices, liqueurs, ice creams, and jellies [5].

Scarce scientific studies have characterized araçá-boi, abiu grande, araticum, biri-biri and yellow mangosteen fruits. The bioactive compounds and total antioxidant activity have been determined in abiu [6,7], cowberry [3,7,8], biri-biri [9], araticum [10,11] and yellow mangosteen [3]. Araçá-boi is characterized by its high content of phenolic compounds, flavonoids and antioxidant activity [1], as well as carotenoids [3] and antimutagenic and antigenotoxic activity [12]. Araticum, besides having a high carotenoid content, is an excellent source of vitamins C and A [13], and a study by Virgolin et al. [3] showed a high content of yellow flavonoids (flavonones and flavonols) in yellow mangosteen.

Numerous studies have shown that fruit consumption is associated with a chronic non-communicable diseases (NCDs) risk reduction [14,15,16]. NCDs, including cardiovascular, neoplastic, ischemic cerebrovascular, and chronic respiratory diseases, as well as diabetes, represent a major public health challenge for the next decade [17]. According to Chaabane et al. (2012) [18], metabolic dysfunctions and oxidative states favor the development of these diseases. Therefore, a diet rich in bioactive compounds which exhibit antioxidant activity, such as ascorbic acid, phenolic compounds, and carotenoids, present in fruits, is capable of inhibiting oxidation processes in the body, therefore representing a promising alternative in NCDs prevention and treatment [19,20,21].

Bioactive compounds from fruits have been shown to regulate prostaglandin synthesis and cholesterol synthesis and absorption, to reduce platelet aggregation, as well as lowering blood pressure [22]. High fruit consumption is also associated with changes in specific antioxidant markers or early indicators associated with the risk of pathologies, including cholesterol oxidation products, plasma antioxidant capacity, total circulating glucose, and body weight [23].

It is believed that the increase in fruit consumption derives from the awareness of its benefits, thus additional experimental and clinical studies that scientifically demonstrate, through reliable biomarkers, the proposed benefits should be conducted [24]. It is noteworthy that araçá-boi, abiu grande, araticum, biri-biri, and yellow mangosteen are relatively unknown fruits. Thus, it is also important to evaluate their safety for consumption.

There are scarce studies in the literature that correlate the consumption of Amazonian fruits with biological parameters. Thus, Amazonian fruits offer new perspectives for functional foods characterization which, in addition to basic nutritional functions, produce metabolic and/or physiological beneficial effects to health when consumed as part of the regular diet [25].

Given the above, considering the effort in the field of public health to discover alternative sources to reduce the burden of NCDs without presenting toxicity, and that there are few studies conducted in Brazil assessing beneficial potential of Amazonian fruits, the objective of this study was to analyze the effects of consuming the pulp of Amazonian fruits (araçá-boi, abiu grande, araticum, biri-biri, and yellow mangosteen) on hematological, metabolic, liver, and renal function parameters in rats. We also aimed to analyze the chemical composition of these pulps to promote health and quality of life in the population and sustainability in the Amazonian region.

## 2. Material and Methods

### 2.1. Raw Material

The pulps of araçá-boi, abiu grande, araticum, biri-biri, and yellow mangosteen were acquired in the frozen form as 100 g packages from an agro-industry company (latitude: 11°26′19″ S and longitude: 61°26′50″ W) in the city of Cacoal, RO, in February 2016. The pulps were stored in a freezer at −18 °C until analysis.

### 2.2. Physicochemical Analysis

Total soluble solids, total titratable acidity, reducing sugars, non-reducing sugars, total sugars, and pH of the fruit pulps were determined using methods described by the Association of Official Analytical Chemists (AOAC), 2005 [26]. The results were expressed as follows: total soluble solids as degrees brix (°Bx), total titratable acidity as g citric acid/100 g pulp, and reducing sugars, non-reducing sugars, and total sugars as g glucose/100 g pulp. Determination of the ascorbic acid content of fruit pulps was based on the oxidation of ascorbic acid by 2,6-dichlorophenolindophenol reagent [26,27] and the result was expressed as mg ascorbic acid/100 g pulp. The moisture content, ash, lipids, and proteins (total nitrogen by the Kjeldahl method and conversion factor of 6.25) in the fruit pulps were evaluated using previously established methods [26].

Extraction of total phenolic compounds was performed using a method described by Macoris et al. (2012) [28] and their content was determined using the Folin-Ciocalteu colorimetric method [29]. Absorbance readings were performed at 720 nm on a Beckman DU-640 spectrophotometer (Fullerton, CA, USA). Quantification was performed using a calibration curve obtained using standard gallic acid (Sigma Aldrich, St. Louis, MO, USA) solutions (72 to 200 μg/mL) and the results were expressed as mg of gallic acid equivalent (GAE)/100 g pulp.

Total antioxidant activity of fruit pulp was determined using DPPH (Sigma Aldrich, St. Louis, MO, USA) free radical scavenging [4]. The absorbance reading was performed at 515 nm and methanol was used as a blank to calibrate the Beckman DU-640 spectrophotometer (Fullerton, CA, USA). Quantification was performed using a calibration curve obtained using standard Trolox (Sigma Aldrich, St. Louis, MO, USA) solutions (200 to 800 μM). The results were expressed as μmol of Trolox/mg of pulp. Among the antioxidant activity methods (ABTS, DPPH, FRAP, and oxygen radical absorbance capacity (ORAC)), the DPPH method proved to be more efficient for the fruits analyzed in this study according to a study previously carried out by Virgolin et al. [3].

All analyses were performed in triplicates at the food analysis laboratory of the Food Engineering Department, IBILCE, São José do Rio Preto.

### 2.3. Biological Assays

After obtaining the approval for the study from the Ethics Committee in the Use of Animals (CEUA) of the Federal University of Rondônia Foundation—UNIR (Protocol number: PP 020/2014) the entire animal experimentation protocol was carried out according to the rules established by the Brazilian College of Experimentation on Animals, Federal University of São Paulo, World Health Organization and Society for Neuroscience [30]. 

The biological assays were performed using 60 Wistar rats (*Rattus norvegicus*) born on the same day acquired from the Federal University of Mato Grosso (UFMT). Newly weaned male rats with masses between 90 g and 100 g were separated and placed in polypropylene cages with a stainless steel lid (maximum of two rats per cage) in a controlled environment. The rats were kept in 12-h dark/12-h light cycles at 23  ±  1 °C and 55% relative humidity. The animals received standard diet and drinking water ad libitum for seven days. After this period, they were divided into groups of ten animals to form the six treatment groups (T): T1—Control, T2—Araçá-Boi, T3—Abiu-Grande, T4—Araticum, T5—Biri-biri, and T6—Yellow Mangosteen.

The experiment was conducted from March 16 to April 15 of 2016, lasting 30 days (sub-chronic effect). In this period, the animals had free access to water and pelleted feed (Presence^®^) with a balanced nutritional composition (Code No.: 05.7883.40.20; Lot No.: 41EX150844109) for rodents and mortality was monitored. The fruit pulp was administered once a day to the animals through a syringe attached to a curved needle with a rounded tip that was introduced into the animal’s stomach by gastric intubation (gavage), so that an exact amount was administered [31]. The amount administered was 0.3214 mL of pulp/100 g of body mass, which is equivalent to the intake of 500 mL of juice (prepared with 45% pulp) by an adult weighing 70 kg. The animals were weighed daily and the values obtained were recorded to calculate the pulp dose to be administered. After the experimental period, intraperitoneal anesthesia was performed using pentobarbital sodium (50 mg/kg) (Abbott, IL, USA) [32], followed by laparotomy to collect 1–2 mL of blood sample from the abdominal aorta and euthanasia by severing the abdominal aorta [33,34,35].

#### 2.3.1. Hematological Analysis

The hematocrit percentage was determined using the microhematocrit technique (Quimis^®^ microhematocrit centrifuge, São Paulo, Brazil). 

Platelet count was performed via a blood smear stained with hematoxylin and eosin (Renylab, Minas Gerais, Brazil) and analyzed using light microscopy (monochrome microscope Primo Star ZEISS, Oberkochen, Germany) with an immersion objective (100× magnification). The counting was performed in 10 random fields. The mean platelet number per field was then multiplied by 20,000 (correction factor for monochrome microscopes).

Total leukocyte count was performed on 0.4 mL of Turk’s solution (Renylab, Minas Gerais, Brasil) added to 20 uL of whole blood (1:21 dilution) in a test tube. The count was then performed using a hemocytometer Olen Neubauer Chamber (Forlabexpress, Rio de Janeiro, Brazil) in the four quadrants of the chamber, and the calculation was performed according to the following equation:$$\mathrm{Leukocytes}\;\left({\mathrm{mm}}^{3}\right)\;=\;\frac{\mathrm{Number}\;\mathrm{of}\;\mathrm{leukocytes}\;\times \;21\times 10\;}{4}$$
where 10 is the conversion factor for μL, 21 is the dilution conversion factor, and 4 is the denominator [33].

#### 2.3.2. Biochemical Analysis

The protocols and reference values adopted in biochemical analyzes followed those of similar studies previously carried out with experimental animals [34,35,36]. The collected blood was centrifuged for 10 min at 3500 *g* to obtain the plasma. The analyses were performed by spectrophotometry using an automatic analyzer for biochemical and immunochemical tests Labmax 240^®^ (Labtest, Minas Gerais, Brazil). The liver enzymes, glutamic-oxaloacetic transaminase (TGO) and glutamic-pyruvic transaminase (TGP), were measured by spectrophotometry at 340 nm. Total cholesterol, triacylglycerols, and glucose levels were determined by spectrophotometry at 500 nm, and creatinine and urea were measured by spectrophotometry at 504 nm [37,38]. The results were expressed as mg/dl.

#### 2.3.3. Statistical Analysis

The results were tested for normal distribution and homogeneity of variance. When the conditions for applying parametric statistical tests to compare the means were met, the comparison between the results was performed for independent samples using analysis of variance (ANOVA) followed by Tukey’s test at a significance level of 5%. In addition, marginally significant results at the 10% significance level (*p* < 0.10), were also indicated. The Kruskal–Wallis nonparametric statistical test was used in the set of results for which normal distribution and, especially, homogeneity of variance was not observed. The results were expressed as mean ± standard deviation. Statistical analyses were performed using the Statistica 12.0 software (TIBCO, PALO ALTO, CA, USA).

## 3. Results and Discussion

The results of the physicochemical analyses are presented in Table 1. The pulps of abiu and araçá-boi had the highest (*p* ≤ 0.05) and lowest levels of soluble solids, respectively. Among the various fruit components, total soluble solids (°Bx) play a key role in determining fruit quality due to their influence on the thermophysical, chemical, and biological properties of the fruit [39].

The pulp of araçá-boi had the highest level of total titratable acidity (*p* ≤ 0.05), indicating the presence of citric acid, the chief organic acid found in some fruits. Biri-biri pulp had the highest level of ascorbic acid (*p* ≤ 0.05). The levels of ascorbic acid in fruit pulps can be lower than that in fresh fruits because this compound is degraded during food preparation and storage, due to the action of light, temperature, high pH, metal ions (Cu^+2^ and Fe^+3^), reactive oxygen species, and humidity [40]. Smirnoff and Wheeler (2000) [41] suggest that the immediate precursors of ascorbic acid biosynthesis are D-glucosone and L-sorbosone, explaining the low levels of reducing, non-reducing, and total sugars in the araçá-boi and biri-biri pulps, and high levels in the abiu and yellow mangosteen pulps. Moisture content was higher than 80% in all pulps. However, it was relatively lower in the abiu pulp (*p* ≤ 0.05), which is explained by the high level of soluble solids in it (Table 1).

Araticum pulp had the highest ash content (*p* ≤ 0.05) of all pulps. According to Damiani et al. (2011) [5], the predominant mineral in araticum pulp is magnesium (350 mg/kg), followed by phosphorus (220 mg/kg). There are several factors that interfere with the mineral content of fruits, as confirmed in this study, such as variety, stage of maturation, soil type and conditions, fertilization, irrigation, and temperature [42]. Additionally, all fruits had low levels of lipids (≤0.28%) and proteins (≤4.60%).

The araticum, yellow mangosteen, and araçá-boi pulps had the highest contents of total phenolic compounds (258.04, 204.0, and 144.0 g EAG/100 g, respectively; *p* ≤ 0.05), and were thus classified as having medium polyphenol content (100–500 mg EAG/100 g) whereas the other pulps were classified as having low polyphenol content (<100 mg EAG/100 g) according to classification proposed by Vasco, Ruales and Kamal-Eldin (2008) [43].

The pulp of araticum and araçá-boi had the highest total antioxidant activity values (6.27 and 6.09 μmol of Trolox/g, respectively; *p* ≤ 0.05), higher values than those found by Garzón et al. (2012) [8] and Vinholes et al. (2017) [44] that reported total antioxidant activity for araçá pulp of 0.8 μmol and 3.34 μmol of Trolox/g, respectively, using the DPPH method. Souza et al. (2012) [10] reported higher values for antioxidant activity (131.58 μmol of Trolox/g) and total phenolic compounds (739.37 mg EAG/100 g) in araticum from Cerrado; additionally, their values for ascorbic acid and β-carotene levels were 59.05 mg/100 g and 0.57 mg/100 g, respectively. Antioxidant content in fruits is influenced by genetic factors, edaphoclimatic conditions, stage of development, and cultivation system [28].

The results obtained in the physical-chemical characterization of the pulps have driven us to deepen our research on the physiological and metabolic effects beneficial to the health of these Amazonian fruit pulps (Appendix A). This is because, in addition to the functional potential, these fruit pulps could help in prevention and treatment of some diseases [45]. Figure 1 depicts the results of the experimental tests of pulp consumption and its physiological effects on the Wistar rats.

Figure 1A shows that the consumption of araticum pulp (T4) marginally increased (*p* ≤ 0.10) the hematocrit percentage (0.46%) in experimental groups relative to the control group (0.42%) of rats. It may be possible that the increase in hematocrit is due to the antioxidant activity of phenolic compounds found in fruits [46]. According to a study by Lage (2014) [47] the araticum pulp contain mainly flavonoids, such as O-glycosides quercetin and peltatoside, aporphine alkaloid, norstephalagine, and epicatechin, that are the compounds that protect normal cells from the toxic effect of highly reactive oxygen concentration generated during cellular metabolism [48,49]. Although a study conducted by Xu et al. (2017) [50] demonstrated that the ethanolic extract of *Garcinia xanthochymus* increased the activities of antioxidant enzymes (superoxide dismutase, catalase, and heme oxygenase-1) and decreased reactive oxygen species in Pheochromacytoma (PC12) cells in rats, this result differs from that found in our study, where no significant increase in hematocrit was observed after the consumption of yellow mangosteen pulp.

With regard to total leukocytes (Figure 1B), which are cells that participate in the inflammatory process, the results suggest that the consumption of abiu and araticum pulps promoted a significant reduction relative to the control group (4166.67 mm^3^, 4365.00 mm^3^ and 7670.00 mm^3^ respectively, *p* ≤ 0.05). A previous study conducted by Rocha et al. (2016) [51] with ethanolic extract of araticum demonstrated that oral treatment with 300 mg/kg of extract significantly inhibited the formation of carrageenan-induced edema (*p* ≤ 0.05) and decreased the number of leukocytes (*p* ≤ 0.05) in the tested animals, indicating that the extract has chemopreventive and anti-inflammatory potential due to the presence of kaempferol. Similar results were reported by Meira et al. (2014) [52], where the administration of ethanolic extract of abiu leaves in rats had an effect against inflammatory pain and anti-hypersensitive action.

Fruit pulps generally contain compounds with functional potential, such as vitamins, carotenoids, flavonoids, steroids, and fibers (mainly soluble fiber and pectin) [15], that improve glycemic control and have been shown to be effective as adjuvants in diabetes treatment [53]. As shown in Figure 1C, the intake of fruit pulp did not cause changes in blood glucose in the tested rats as all animals had normal fasting blood glucose. However, the consumption of araticum (88.10 mg/dL) and biri-biri (89.83 mg/dL) pulps marginally reduced their blood glucose (*p* ≤ 0.10).

Previous studies conducted by Anitha, Geetha, and Lakshmi (2011) [54] and Pushparaj, Tanb, and Tana (2001) [55] demonstrated that an ethanolic extract of biri-biri leaves had hypoglycemic activity in rats by decreasing glucose-6-phosphatase activity without affecting cytochrome P_450_. The aqueous extracts of yellow mangosteen and abiu fruits, that was previously studied, contain bioflavonoids, including GB2a glucoside, GB2a, and fukugetinde [56]; additionally, according to Souza et al. (2012) [10], they strongly inhibit the activity of alpha-amylase and alpha-glucosidase. According to previous studies, the antihyperglycemic effect via the inhibition of alpha-glucosidase and carbohydrate metabolism enzymes of glycosylated quercetin derivatives, are found at high levels in araçá-boi pulp [57,58]. However, these results are not in accordance with our study, that observed no significant reduction in blood glucose (*p* > 0.05) after the consumption of abiu, yellow mangosteen, and araçá-boi pulps by rats.

The consumption of fruits and their phytochemical components, especially polyphenols and fibers is related to the reduction or control of blood cholesterol and triglycerides levels [59]. This statement is partially in agreement with our results (Figure 1D) that demonstrate that, with the exception of araticum pulp (T4), the consumption of Amazonian fruit pulps causes a marginal reduction in plasma cholesterol and a marginal increase in triglycerides in rats, although not significantly (*p* ≤ 0.10). The increase in triglycerides may be associated with the fact that most fruits have high concentrations of simple sugars, in particular fructose [60], a fact that may have modulated the metabolism of rats that consume the pulps of Amazonian fruits. As already reported, a significant difference (*p* ≤ 0.05) in triglycerides was observed between the control group (61.44 mg/dL) and the araticum pulp group (99.50 mg/dL). This could be due to the low ascorbic acid content in the fruit pulp (3.47 mg of ascorbic acid/100 g), in addition to the content of total sugars (5.6 g of glucose/100 g), parameters analyzed in our study, and mineral iron (0.43 g/100 g), a parameter analyzed in another study [61]. Despite the beneficial effects of minerals, high iron content can cause lipid peroxidation via the formation of free radicals and is thus a risk factor for atherosclerosis [62]. 

One way to assess liver function is to perform routine biochemical testing of the serum liver enzymes, such as cytoplasmic TGO and TGP, as well as serum levels of creatinine and urea to assess renal function [63,64]. As shown in Figure 1E, serum levels of TGP in the different experimental groups were not significantly different (*p* > 0.05) when compared to the control group of rats. Although previous study show low in vitro cytotoxicity of biri-biri methanolic extract [65], in our study the regular consumption of its pulp for 30 days significantly (*p* ≤ 0.05) increased serum TGO levels in rats (116.83 U/L), compared to the control group (98.00 U/L). It is noteworthy that three Wistar rats from the group that consumed biri-biri pulp died within the 30-day experiment. According to the literature, elevated levels of TGO and TGP induce an increase in bilirubins, in particular direct bilirubin, alkaline phosphatase, and gamma-glutamyl transferase, which indicate liver toxicity [66].

Creatinine is a catabolite of muscle metabolism of creatine-phosphate; it is primarily eliminated via glomerular filtration, with a small amount being excreted via tubular secretion. It has been used as a biomarker of chronic kidney disease and acute kidney injury [67]. No significant difference (*p* > 0.05) was observed in creatinine after the consumption of the fruit pulps in the experimental groups relative to the control group (Figure 1F).

Urea is a waste product excreted in the urine. However, it also plays an important role in human physiology by assisting the function of nephrons, which are microscopic structures capable of eliminating metabolic waste from the blood, maintaining the hydroelectrolytic and acid–base balance, controlling the amount of fluids, regulating blood pressure, secreting hormones, and producing urine [68]. Herein, the results of urea test (Figure 1G) demonstrated that only the experimental group that consumed biri-biri pulp had a marginal increase (*p* ≤ 0.10) in urea levels (28.17 mg/dL). The result obtained in this study is similar to the findings of Bakul et al. (2013) [69]. Biri-biri has a high oxalic acid content and its consumption is linked to a high risk of developing acute renal failure due to deposition of calcium oxalate crystals in the renal tubules. Paschoalin et al., (2014) [70] reported in a case study, acute kidney injury associated with lumbar pain, hiccups, and diarrhea in a 50-year-old hypertensive patient with normal kidney function who consumed a large amount of biri-biri juice after fasting to treat hypertension. Thus, between the evaluated Amazonian fruit pulps by us, only the intake of biri-biri pulp showed risks of liver and kidney dysfunction in rats.

## 4. Conclusions

Among the Amazonian fruits studied, Araticum pulp had higher ash content, total phenolic compounds, and antioxidant activity than the other analyzed pulps. Moreover, araticum pulp consumption by rats marginally increased the hematocrit percentage (*p* ≤ 0.10), promoted a significant reduction in total leukocytes (*p* ≤ 0.05), and a marginal reduction on blood glucose (*p* ≤ 0.10).

Abiu pulp had the highest levels of soluble solids, total and reducing sugars, and pH, and the lowest level of moisture. Furthermore, abiu consumption promoted a significant reduction (*p* ≤ 0.05) in total leukocytes of rats.

Biri-biri pulp had the highest levels of ascorbic acid and total titratable acidity and a low pH and its consumption promoted a marginal reduction on blood glucose (*p* ≤ 0.10). However, attention should be paid to its consumption due the risks of liver and kidney dysfunction observed.

Additionally, as all fruit pulps had low lipid and protein contents, the consumption of Amazonian fruit pulps caused a marginal reduction in plasma cholesterol, but a marginal increase in triglycerides in rats, although not significantly (*p* ≤ 0.10). Exception should be made to Araticum consumption, where the increase in triglycerides was significant (*p* ≤ 0.05).

These results generate knowledge mainly with regard to the chemical and nutritional composition and effects of the intake of native/cultivated fruits in the Amazon biome. However, more researches are needed to generate more knowledge about these promising Amazonian fruits so as to promote the native fruit production, in addition to promoting health in the population and sustainability in the Amazon region.

## Figures and Tables

**Figure 1 nutrients-13-01484-f001:**
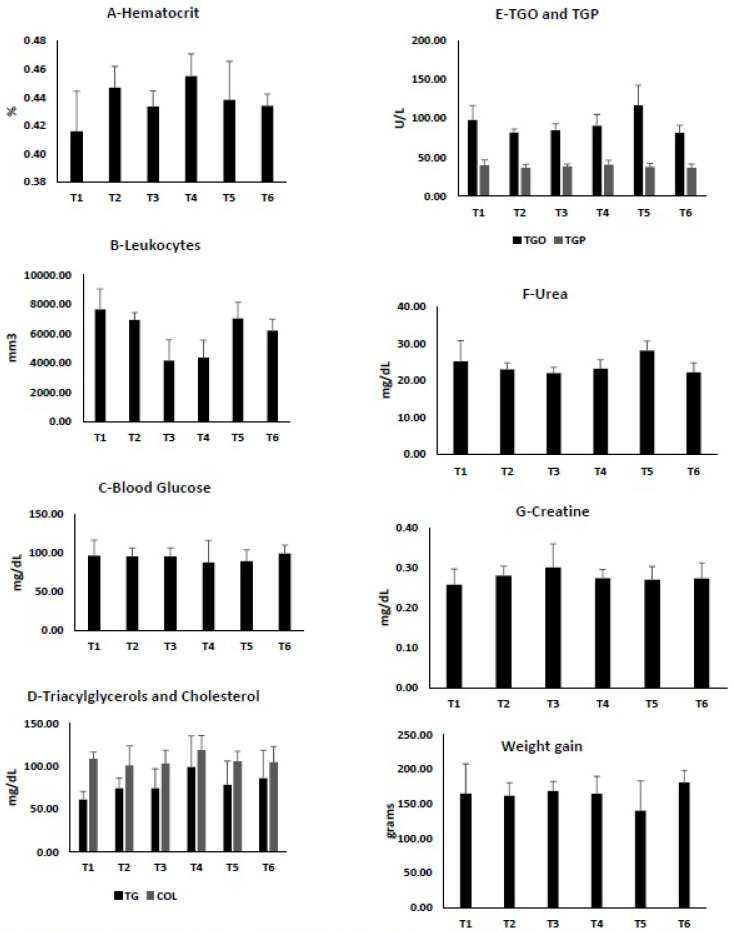
Biochemical results after Amazonian pulp fruits consumption in Wistar rats (*n* = 10 rats/group), T1—Control; T2—Araçá-Boi; T3—Abiu-grande; T4—Araticum; T5—Biri-biri; T6—Yellow mangosteen; TGO—glutamic-oxaloacetic transaminase; TGP—glutamic-pyruvic transaminase; TG—triacylglycerol and Col—Cholesterol. Mean values and standard deviation are presented.

**Table 1 nutrients-13-01484-t001:** Physicochemical composition of fruit pulps from the Brazilian Amazon biome.

Parameters ^1^	Araçá-Boi *(Eugenia stipitata)*	Abiu Grande *(Pouteria caimito)*	Araticum *(Annona crassiflora)*	Biri-Biri (*Averrhoa bilimbi* L.)	Yellow Mangosteen (*Garcinia xanthochymus*)
Total soluble solids (°Brix)	3.63 ^e^ ± 0.12	17.50 ^a^ ± 0.00	8.37 ^b^ ± 0.23	4.83 ^d^ ± 0.29	7.00 ^c^ ± 0.00
Total titratable acidity (g citric acid/100 g)	2.34 ^a^ ± 0.05	0.10 ^e^ ± 0.01	0.66 ^d^ ± 0.00	1.02 ^c^ ± 0.01	1.41 ^b^ ± 0.01
Ascorbic acid (mg ascorbic acid/100 g)	13.71 ^b^ ± 0.02	3.11 ^d^ ± 0.31	3.47 ^d^ ± 0.84	20.23 ^a^ ± 0.22	5.60 ^c^ ± 0.62
Reducing sugars (g glucose/100 g)	0.27 ^d^ ± 0.03	5.78 ^a^ ± 0.31	5.18 ^b^ ± 0.04	2.75 ^c^ ± 0.09	5.86 ^a^ ± 0.30
Non-reducing sugars (g glucose/100 g)	0.57 ^b^ ± 0.07	3.40 ^a^ ± 0.24	0.43 ^b^ ± 0.16	0.43 ^b^ ± 0.22	0.65 ^b^ ± 0.020
Total sugars (g glucose/100 g)	0.84 ^e^ ± 0.02	9.19 ^a^ ± 0.32	5.60 ^c^ ± 0.14	3.14 ^d^ ± 0.12	6.51 ^b^ ± 0.50
pH	2.81 ^d^ ± 0.01	5.84 ^a^ ± 0.01	4.06 ^b^ ± 0.01	2.15 ^e^ ± 0.01	2.97 ^c^ ± 0.00
Moisture (%)	95.35 ^b^ ± 0.02	81.65 ^e^ ± 0.03	89.25 ^d^ ± 0.14	96.23 ^a^ ± 0.03	91.65 ^c^ ± 0.05
Ashes (%)	0.19 ^b^ ± 0.01	0.32 ^b^ ± 0.02	2.32 ^b^ ± 0.30	0.20 ^a^ ± 0.07	0.31 ^b^ ± 0.00
Lipids (%)	0.07 ^c^ ± 0.02	0.13 ^b^ ± 0.01	0.09 ^c^ ± 0.00	0.06 ^c^ ± 0.02	0.28 ^a^ ± 0.01
Protein (%)	2.30 ^b^ ± 0.01	4.60 ^a^ ± 0.31	2.52 ^b^ ± 0.11	1.27 ^c^ ± 0.12	1.19 ^c^ ± 0.00
Total phenolic compounds (mg EAG/100 g fruit)	144.0 ^c^ ± 0.06	60.0 ^d^ ± 0.015	258.04 ^a^ ± 0.036	30.0 ^e^ ± 0.005	204.0 ^b^ ± 0.03
Antioxidant activity (µmol Trolox/100 g)	6.09 ^a^ ± 0.29	2.89 ^c^ ± 0.00	6.27 ^a^ ± 0.06	0.62 ^d^ ± 0.00	4.29 ^b^ ± 0.26

^1^ Mean ± standard deviation. Means followed by the same letters in a line do not differ significantly (*p* ≤ 0.05) by the one-way ANOVA test followed by post-hoc Tukey test.

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
