# Peer review of "Physicochemical Properties and Effects of Fruit Pulps from the Amazon Biome on Physiological Parameters in Rats"

_nutrients, 2021, doi:10.3390/nu13051484_

Round 1

Reviewer 1 Report

The authors adequately incorporated the comments and improved significantly the quality of the manuscript.

I only have a few minor corrections:

L224, Insert ‘the’ in front of ‘compounds’ and change ‘cell’ to ‘cells’ and ‘high’ to ‘highly’.

Figure 1-D, TG and Col are not defined, please state them in caption.

L331, shouldn’t p value be > 0.05?

Author Response

We would like to thank for the appointments, they were very important for the improvement of our paper, and inform that each point will be answered below.

The authors adequately incorporated the comments and improved significantly the quality of the manuscript.

I only have a few minor corrections:

L224, Insert ‘the’ in front of ‘compounds’ and change ‘cell’ to ‘cells’ and ‘high’ to ‘highly’.

Figure 1-D, TG and Col are not defined, please state them in caption.

L331, shouldn’t value be > 0.05?

Point 1: L224, Insert ‘the’ in front of ‘compounds’ and change ‘cell’ to ‘cells’ and ‘high’ to ‘highly’.

Response 1: It was done.

Point 2: Figure 1-D, TG and Col are not defined, please state them in caption.

Response 2: It was done.

Point 3: L331, shouldn’t p value be > 0.05?

Response 3: Yes, and it was correct.  

Reviewer 2 Report

The introduction and objectives have been improved, although some aspects of methods remains inconsistent and some errors persist. In addition, the discussion remains quite confusing. The presentation of results and comparison with other studies is not clear and there is not well contextualized. In general, you make relatively excessive conclusions, as you do not have sufficient/adequate data or results for some of the associations presented.
Detailed comments:
Lines 44-45: The species names should be in italic form, likwise the abstract.
Line 53: What is yellow flavonoids?
Line 69-70: What does novel foods mean in this sentence?: “It is noteworthy that because they are novel foods, it is also important to evaluate their safety for consumption”.
Line 191: Table 1. You should put “ácido cítrico” in english language and “yellow mangosteen” is not in italic form.
Line 209: You should not make a paragraph, because you continue to mention the issue related to antioxidant activity.
Line 219-221: Please, put the type of letter in acordance to the Nutrients journal template.
Line 223-224: this seems that you identified the main flavonoids in the araticum pulp, which is not true. You should rewrite the sentence in order to clarify that the TPC found can be associated with flavonoids described in the literature (examples and reference).
Line 230: Again, in this sentence you should clarify that these parameters have been evaluated in previous studies, with good results. “These parameters were used previously...”.
Line 236: In the Fig 1. you should put a title in each graph in accordance: A-Hematocrit, B-Leukocytes, C- Blood glucose, D-Triacylglycerols and Cholesterol,  F-Urea, G-Creatine, H-Body Weight
Lines 241 to 245: This paragraph sould be before the Fig.1, because the content of the text is related to the previous paragraph.
Lines 247 – 249: the leucocyte counts should be appear in the final of the sentence, such as (4166.67, 4365.00 and 7670.00 mm3 repectively, p ≤ 0.05), improving clarity for readears.
Line 250 and 251: Again, you should clarify the information. The p value should apear after the respective parameter: "inhibited the formation of carrageenan-induced edema (p ≤ 0.05) and decreased the number of leukocytes (p ≤ 0.05).
Line 253 to 257: The results of these studies are in accordance or related to your results? In these studies the authors showed different activities of araticum (inhibition of digestive enzymes, antibacterial and hepatoprotective), as well as the phenolic coumpounds, that you did´t studied. You should use this information in the introduction part, where you refer biological proprieties of araticum (line 49). 
Line 258 – 264: Again, you dont understand my comment. The results of these studies are in accordance or related to your results? In point 15, I advised that you should perform other analyzes to assess the anti-inflammatory activity, such as the inhibition of pro-inflammatory cytokines, intercellular adhesion molecules and enzymes (e.g., iNOS, COX-2, 5-LOX), or transcription factors (e.g. NF-κB), and not add the results showed by others, like you refered “The affirmation of the fruit's anti-inflammatory activity was given by other references cited in the text”. In addition, the new rewritten text refers to A. crassiflora. Is that a fruit related to the studied fruits? If not, you can´t introduce this
Lines 265 - 270: Again, there is a lack of context. How do the results found in these studies relate to your results? 
Lines 271 – 276: Again, these sentences must also be contextualized and accompanied by bibliographic references.
Lines 286 – 287: Did you study the antibacterial and cytotoxic activity of the species? So why do you mention data on these effects in the discussion? This could be mentioned in a general way in the introduction, if it had been evaluated in the same species that you studied ...
Line 288 - 295: You should contextualize the results of other studies,  as “In a previous study...” or “it was previously studied...”, “are/are not in accordance with our resulsts”.
Lines 296, 300, 305 and 308: Again, you should contextualize the results of other studies. These paragraphs are related. You should present your results and discuss the main observations, comparing with what has been described and related in other studies, making connections with various aspects, such as the case of dyslipidemia.
Line 333 - 340: This sentence is not clear. It is not easy to understand what are the results of your study and that of the other studies [81][82].
In the Conclusion part, you should explain what can means that "the intake of araticum increased significantly triglycerides". Herein, you should not present results.
The journal´s names should be in abbreviated form according to References rules of the template of Nutrients (Abbreviated Journal Name).

Author Response

We would like to thank for the appointments, they were very important for the improvement of our paper, and inform that each point will be answered below.

Point 1: Lines 44-45: The species names should be in italic form, likwise the abstract.

Response 1: It was done.

Point 2: Line 53: What is yellow flavonoids?

Response 2: Flavones and flavonols are widely distributed in plants as O-glycosides. Flavonols such as quercetin, myricitin and rutin have an OH group at C3, whereas the flavones, (e.g., apigenin, luteolin, and baicalein;) have a hydrogen in that position. The flavonoid content can be obtained using two methods. The first method uses aluminum chloride and is based on the formation of a complex between the aluminum ion, Al (III), and the carbonyl and hydroxyl groups of flavones and flavonols that produce a yellow color.

Point 3: Line 69-70: What does novel foods mean in this sentence?: “It is noteworthy that because they are novel foods, it is also important to evaluate their safety for consumption”.

Response 3: The sentence has been modified to: It is noteworthy that because they are relatively unknown fruits, it is also important to evaluate their safety for consumption.

Point 4: Line 191: Table 1. You should put “ácido cítrico” in english language and “yellow mangosteen” is not in italic form.

Response 4: the modification was carried out

Point 5: Line 209: You should not make a paragraph, because you continue to mention the issue related to antioxidant activity.

Response 5: We join the paragraphs

Point 6: Line 219-221: Please, put the type of letter in acordance to the Nutrients journal template.

Response 6: We format the paragraph.

Point 7: Line 223-224: this seems that you identified the main flavonoids in the araticum pulp, which is not true. You should rewrite the sentence in order to clarify that the TPC found can be associated with flavonoids described in the literature (examples and reference).

Response 7: The paragraph has been redone.

Point 8: Line 230: Again, in this sentence you should clarify that these parameters have been evaluated in previous studies, with good results. “These parameters were used previously...”.

Response 8: The paragraph has been redone.

Point 8: Line 236: In the Fig 1. you should put a title in each graph in accordance: A-Hematocrit, B-Leukocytes, C- Blood glucose, D-Triacylglycerols and Cholesterol,  F-Urea, G-Creatine, H-Body Weight.

Response 8: titles were entered in each graph.

Point 9: Lines 241 to 245: This paragraph sould be before the Fig.1, because the content of the text is related to the previous paragraph.

Response 9: the paragraph has been relocated.

Point 10: Lines 247 – 249: the leucocyte counts should be appear in the final of the sentence, such as (4166.67, 4365.00 and 7670.00 mm3 repectively, p ≤ 0.05), improving clarity for readears.

Response 10: leukocyte counts were reallocated.

Point 11: Line 250 and 251: Again, you should clarify the information. The p value should apear after the respective parameter: "inhibited the formation of carrageenan-induced edema (p ≤ 0.05) and decreased the number of leukocytes (p ≤ 0.05).

Response 11: The p value were reallocated.

Point 12: Line 253 to 257: The results of these studies are in accordance or related to your results? In these studies the authors showed different activities of araticum (inhibition of digestive enzymes, antibacterial and hepatoprotective), as well as the phenolic coumpounds, that you did´t studied. You should use this information in the introduction part, where you refer biological proprieties of araticum (line 49). 

Response 12: The paragraph has been removed.

Point 13: Line 258 – 264: Again, you dont understand my comment. The results of these studies are in accordance or related to your results? In point 15, I advised that you should perform other analyzes to assess the anti-inflammatory activity, such as the inhibition of pro-inflammatory cytokines, intercellular adhesion molecules and enzymes (e.g., iNOS, COX-2, 5-LOX), or transcription factors (e.g. NF-κB), and not add the results showed by others, like you refered “The affirmation of the fruit's anti-inflammatory activity was given by other references cited in the text”. In addition, the new rewritten text refers to A. crassiflora. Is that a fruit related to the studied fruits? If not, you can´t introduce this

Response 13: We understand your comment.

We thought it would be interesting to introduce which parameters are needed to be analyzed to confirm the anti-inflammatory activity of a food. And yes, Annona crassiflora is the scientific name of the araticum, a fruit that was researched in our study. However, the paragraph has been removed.

Point 14: Lines 265 - 270: Again, there is a lack of context. How do the results found in these studies relate to your results? 

Response 14: The paragraph has been removed.

Point 15: Lines 271 – 276: Again, these sentences must also be contextualized and accompanied by bibliographic references.

Response 15: Despite the known decrease in leukocytes in immunosuppressed patients and the increase in serum leukocytes in patients with inflammation, this paragraph was removed because we did not find consistent data from the scientific literature that relates the consumption of these fruits to the parameters mentioned.

Point 15: Lines 286 – 287: Did you study the antibacterial and cytotoxic activity of the species? So why do you mention data on these effects in the discussion? This could be mentioned in a general way in the introduction, if it had been evaluated in the same species that you studied ...

Response 15: The data has been removed.

Point 16: Line 288 - 295: You should contextualize the results of other studies,  as “In a previous study...” or “it was previously studied...”, “are/are not in accordance with our resulsts”.

Response 16: The paragraph has been rewritten.

Point 17: Lines 296, 300, 305 and 308: Again, you should contextualize the results of other studies. These paragraphs are related. You should present your results and discuss the main observations, comparing with what has been described and related in other studies, making connections with various aspects, such as the case of dyslipidemia.

Response 17: The paragraphs has been rewritten.

Point 18: Line 333 - 340: This sentence is not clear. It is not easy to understand what are the results of your study and that of the other studies [81][82].

Response 18: The paragraphs has been rewriten.

Point 19: In the Conclusion part, you should explain what can means that "the intake of araticum increased significantly triglycerides". Herein, you should not present results.

The journal´s names should be in abbreviated form according to References rules of the template of Nutrients (Abbreviated Journal Name).

Response 19: The paragraphs has been rewriten and the journal´s names have been corrected.

Reviewer 3 Report

I have liked the effort you have made to improve the publication. Especially since a lot of research is needed to generate more knowledge of the native fruits of the Amazon region to help all the indigenous people who live in that region, not only for their sustainability but also for their health and general well-being. I think that research like this helps to do this and, above all, the best thing will be that in the future these studies are truly transferred for the good of the entire region and its people. 

You have made a better introduction including other similar studies and considerably more references that you did not have before. This has helped to better focus your work. I have also noticed that you have better described the methods and techniques used in the study and their conclusions have been much better supported by their results. 

I encourage you to continue this research in more depth to help get more results that help this region of the Amazon.

Author Response

We would like to thank for the comprehension about this study importance to the Amazonian population health and sustainability.

Reviewer 4 Report

The methodology section needs improvement.

The summary of the basic physicochemical properties of plant material may be a good introduction to further advanced research. The Folin-Ciocalteu colorimetric method should be used for the preliminary and screening determination of phenolic compounds. The reviewer also doubts that the use of only one antioxidant activity test, the specific DPPH test, may be insufficient to conclude on the antioxidant activity.

Conclusions cannot repeat the results. Please indicate the main conclusions, what new research has brought to science, what is their measurable benefit.

The manuscript should be adapted to the requirements of MDPI and its style and quality improved. For example, please change the Spanish words 'fruta', 'de ácido cítrico', 'de glucose' in Table 1 to the language of the paper.

Author Response

We would like to thank for the appointments, they were very important for the improvement of our paper, and inform that each point will be answered below.

Point 1: The methodology section needs improvement.

Response 1: Could you specify which part of the methodology needs improvement? we made several modifications requested by the other reviewers.

Point 2: The summary of the basic physicochemical properties of plant material may be a good introduction to further advanced research.

Response 2: This is exactly our goal, as presented in the conclusion.

 Point 3: The Folin-Ciocalteu colorimetric method should be used for the preliminary and screening determination of phenolic compounds. The reviewer also doubts that the use of only one antioxidant activity test, the specific DPPH test, may be insufficient to conclude on the antioxidant activity.

Response 3: The ABTS, DPPH, FRAP, and orygen radical absorbance capacity (ORAC) methods used to determine the antioxidant activity are most commonly used in fruits. It is recommend using more than one method (and preferably all of them), in order to acquire more complete information on the antioxidant capacity of foods. This variety of testing allows researchers to consider the advantages or disadvantages of each method, as well as each method’s acceptability.

In a previous study by Virgolin, Seixas and Janzantti (2017) it was observed that the DPPH method was the most efficient to evaluate the antioxidant capacity of the fruit pulps analysed in this study. Therefore, this method was used for our study and that information was inserted in the methodology.

Virgolin, L.B.; Seixas, F.R.F.; Janzantti, N.S. Composition, content of bioactive compounds, and antioxidant activity of fruit pulps from the Brazilian Amazon biome. Pesq. agropec. bras. 2017, 52, 10, 933-941; DOI: 10.1590/S0100-204X2017001000013.

Point 4: Conclusions cannot repeat the results. Please indicate the main conclusions, what new research has brought to science, what is their measurable benefit.

Response 4: We corrected some points of the conclusion.  

Point 5: The manuscript should be adapted to the requirements of MDPI and its style and quality improved. For example, please change the Spanish words 'fruta', 'de ácido cítrico', 'de glucose' in Table 1 to the language of the paper.

Response 5: Yes, and it was correct.  

Round 2

Reviewer 2 Report

The introduction and objectives have been improved, although the discussion remains quite confusing. The presentation of results and comparison with other studies is not clear and there is not well contextualized. In general, you make relatively excessive conclusions, and do not present the results in a clear form. You refer values obtained in other studies that used different parameters, as well as reference values in humans, without a clear discussion of the results obtained.

You still do not discuss or refer to the mortality seen in the study, it should be explain in a clear form in the manuscript.

You also continue to present reference ranges, relative to values for humans that you refer to as "normal" (e.g. for urea), when the study was developed in rats, so these comparisons are wrong or excessive.

Detailed comments:

Lines 70 and 71: What does "they" is refering to in this sentence?: "It is noteworthy that because they are relatively unknown fruits..."

In the methodology part, line 133 states that you monitored mortality, but in the results you say nothing about it.

Lines 140 and 147: Why did you underline some words?

Line 224: Why you conclude that may be possible that the increase in hematocrit is due to the antioxidant activity of araticum pulp? The studies that you refer below, do not mention the hetatocrit.

Lines 225 to 236: The description of results from other studies, corresponding to parameters that were not evaluated in your study, remains quite excessive.

Line 267: “alpha-amylase (IC50 13.6 267 μg/mL) and alpha-glucosidase (IC50 2.58 μg/mL)”, in my opinion, it makes no sense to present numerical results from other studies without being directly related to the results obtained in the present study. In this case, the study presented the IC50 values, but you not explain what is means.

279-280: You should explain the main implication of this result "a significant difference (p ≤ 0.05) in triglycerides was observed between the control group (61.44 mg/dL) and the araticum pulp group (99.50 mg/dL)”.

Lines 286 – 292: To assess liver function, you determined only GOT and GPT, these changes alone do not allow conclusions to be drawn about liver toxicity. The same is true for the conclusions on renal toxicity, where you evaluated only the plasma concentrations of urea and creatinine. These parameters are used, together with others, to estimate liver and kidney function, and not toxicity, as you refer between lines 288 and 290, referencing these statements.

Line 294-295: Why did you present herein “the normal range of serum TGP”? You can relate the value obtained only to the control. What is the sense of presenting human values? It gets confusing.

Line 300 and 314: “TGO values (reference value for TGO = 81.0 U/L [68]” and “The normal range of urea is 20–40 mg/dL [71]”: are these values, reference value for humans too? Can you compare the values obtained with human values? What is the purpose of presenting the human reference values?

In the Conclusion part, you should explain what can mean the following observation "the intake of araticum increased significantly triglycerides". You must underline in a clear form the main results and conclusions of the study, and this is not evident.

Author Response

We would like to thank for the detailed review and appointments made, they were very important for the improvement of our paper, and inform that each point will be answered below.

Point 1: Lines 70 and 71: What does "they" is refering to in this sentence?: "It is noteworthy that because they are relatively unknown fruits..."

Response 1: Your suggestion was accepted and sentence was corrected.

Line 70: It is noteworthy the araçá-boi, abiu grande, araticum, biri-biri  and yellow mangosteen are relatively unknown fruits, thus, it is also important to evaluate their safety for consumption.

Point 2: In the methodology part, line 133 states that you monitored mortality, but in the results you say nothing about it?

Response 2: Mortality was monitored so that in the groups where there was no mortality, it was not reported and in which ones there was mortality, this was reported in the paper, as posted below:

Line 286: It is noteworthy that three Wistar rats from the group that consumed biri-biri pulp died within the 30-day experiment.

Point 3: Lines 140 and 147: Why did you underline some words?

Response 3: The modification was carried out.

Point 4: Line 224: Why you conclude that may be possible that the increase in hematocrit is due to the antioxidant activity of araticum pulp? The studies that you refer below, do not mention the hematocrit.

Response 4: As requested, a new reference that supports the effect of antioxidant activity on hematocrit was included.

Line 223-26: Figure 1-A shows that the consumption of araticum pulp (T4) marginally increased (p≤0.10) the hematocrit percentage (0.46%) in experimental groups relative to the control group (0.42%) of rats.  It may be possible that the increase in hematocrit is due to the antioxidant activity of phenolic compounds found in fruits [47].

Point 5: Lines 225 to 236: The description of results from other studies, corresponding to parameters that were not evaluated in your study, remains quite excessive.

Response 5: We accepted your suggestion and the paragraph has been removed.

Point 6: Line 267: “alpha-amylase (IC50 13.6 267 μg/mL) and alpha-glucosidase (IC50 2.58 μg/mL)”, in my opinion, it makes no sense to present numerical results from other studies without being directly related to the results obtained in the present study. In this case, the study presented the IC50 values, but you not explain what is means.

Response 6: We accepted your suggestion and the values ​​were taken from the paragraph.

Point 7: 279-280: You should explain the main implication of this result "a significant difference (p ≤ 0.05) in triglycerides was observed between the control group (61.44 mg/dL) and the araticum pulp group (99.50 mg/dL)”.

Response 7: We accepted your suggestion, and, as posted below, talked about the biochemical mechanism involved and inserted one more reference to support the sentence.

Line 271: The increase in triglycerides may be associated with the fact that most fruits have high concentrations of simple sugars in particular, fructose [61], fact that may have modulated the metabolism of rats that intake the pulps of Amazonian fruits.

Point 8: Lines 286 – 292: To assess liver function, you determined only GOT and GPT, these changes alone do not allow conclusions to be drawn about liver toxicity. The same is true for the conclusions on renal toxicity, where you evaluated only the plasma concentrations of urea and creatinine. These parameters are used, together with others, to estimate liver and kidney function, and not toxicity, as you refer between lines 288 and 290, referencing these statements.

Line 294-295: Why did you present herein “the normal range of serum TGP”? You can relate the value obtained only to the control. What is the sense of presenting human values? It gets confusing.

Line 300 and 314: “TGO values (reference value for TGO = 81.0 U/L [68]” and “The normal range of urea is 20–40 mg/dL [71]”: are these values, reference value for humans too? Can you compare the values obtained with human values? What is the purpose of presenting the human reference values?

Response 8: We accepted your suggestion and the reference values in humans were removed and the results of the experimental groups were only discussed in comparison with the control group.

As the study had no emphasis on toxicology, we removed the term "toxicity" and changed it to kidney and liver dysfunction.

Point 9: In the Conclusion part, you should explain what can mean the following observation "the intake of araticum increased significantly triglycerides". You must underline in a clear form the main results and conclusions of the study, and this is not evident.

Response 8: As asked, we discussed better mechanism related to triglycerides levels increase in discussion and rewrote conclusion, grouping and highlighting the main results found.

Reviewer 4 Report

The Authors took into account most of the comments and made corrections.
However, a few shortcomings remained, such as 'g de citric acid' and Figure 1 of poor quality (indistinct).
In addition, it would be worth including information on DPPH activity in the discussion, which was included in the reviewer's response, to emphasize the Authors' awareness.

Author Response

Response to Reviewer 4 Comments

We would like to thank for the detailed review and appointments made, they were very important for the improvement of our paper, and inform that each point will be answered below.

Point 1: The Authors took into account most of the comments and made corrections.

However, a few shortcomings remained, such as 'g de citric acid' and Figure 1 of poor quality (indistinct).

Response 1: We fix g citric acid. Regarding the figure, is your suggestion related to the image quality or distinction of the bars in the graphs? We chose to leave the bars in the same colors and nominate them with the experimental groups initials (T1, T2 ...) for a better appearance of the figure. If we added colors, as we have a lot of experimental groups, the figure would not be presentable.

Point 2: In addition, it would be worth including information on DPPH activity in the discussion, which was included in the reviewer's response, to emphasize the Authors' awareness.

Response 2: We understand that this information would fit better in methodology than in the discussion section, as posted below:

Line 112-114: Among the antioxidant activity methods (ABTS, DPPH, FRAP, and oxygen radical absorbance capacity (ORAC), the DPPH method proved to be more efficient for the fruits analyzed in this study according to a study previously carried out by Virgolin et al. [3]

This manuscript is a resubmission of an earlier submission. The following is a list of the peer review reports and author responses from that submission.

Round 1

Reviewer 1 Report

The manuscript describes the physicochemical properties and the effects of fruit pulps on physiological parameters in rats. Although it has a merit in contributing the knowledge to the scientific community, the manuscript requires extensive editing.

  1. Introduction needs more items of literature reviews. L170-176, L186-190 and L235-240 should be in Intro. Add reviews on compounds generally found in fruit pulps and possible effects of the compounds on physiological parameters. NCDs were mentioned, but how these are relevant to the parameters used in the current study is not clearly mentioned.
  2. I encourage to use references in Discussion, but some sections in Results and Discussion are simply literature reviews that should be in Intro as mentioned in 1. For example, once a link between antioxidant activity and high levels of phenolic compounds (L186-190 and L235-240) is mentioned in Intro, your results to support the statement can be discussed.
  3. L96, add details of photoperiod, e.g. 12, 0.5, 11: 0.5: light, dusk, dark, dawn, etc. There should also be marginal temperature and relative humidity settings, e.g. 23 ± 1 °C, 55 ± 1%, etc.
  4. L177-185, these short paragraphs are not coherently written, readers would not be able to understand what the authors try to say.
  5. L197-199, L252, L277 if statistically not significant there shouldn’t be a statement, like ‘marginally increased’, ‘marginally reduced’, ‘marginal increase’. These sentence needs to be revised.
  6. L200, remove ‘a result’.
  7. L266-269, this should be in Introduction.

Reviewer 2 Report

The abstract is not well defined, because does not refer the introduction, methodology and conclusions.

The introduction is poor, lacking the descrition of the specific association between the properties of the amazon fruits under study, and the beneficial effects on health, with support of bibliographic references from related research studies. The introduction is also insufficient in the deepening and describing of hematological, metabolic, liver toxicity, and renal toxicity parameters, evaluated in this study.I suggest the description of your results and the comparison with results found in other articles that used similar methods.

Some parts of work are not well structured and the organization of the manuscript should be corrected. I recommend the uniformity of the abbreviations throughout the manuscript, as well as the methods used. Each abbreviation should be defined at the first time in the text, and then should be used the same abbreviation along the manuscript.

Detailed comments/suggestions:

In line 42, when refer "Thus, amazonian fruits ..." I think that it is better to introduce here the fruits included in the study, descrived after in lines 46 to 49.

In line 64: Please, refer the company name, city and country.

Line 69 and 83: AOAC and DPPH, is not defined these acronyms.

Line 93: the specie name (Rattus norvegicus) should be reffered after “Wistar rats”.

Line 97: Please, indicate the company where you purchased the standard diet.

Line 104: The fruit pulp was administered once a day?

Line 110: Please, indicate the company where you purchased the pentobarbital sodium.

Line 112: Please, indicate the reference of the respective euthanasia protocol.

Line 113: 2.3.1. Hematological analysis: you must indicate bibliographic references of the protocols used, as weel as the companies name, city and country where the materials and equipment were purchased. This must be done throughout the manuscript.

Line 114: You should put in g, instead of rpm.

Line 128: Please, indicate the methods used to determine the parameters referred, and indicate the companies name, city, country, where the materials and equipment were purchased.

Line 143: “Statistica 12.0 software (TIBCO)”: please,  indicate the companies name, city, country, where this software is commercialized.

Throughout the text, particularly in the results and discussion, the numbers of the references used for the studies / authors do not correspond with the numbers in the references. This difficult the interpretation of the results discussion.

Lines 196 to 203: only due to a moderate or slight increase in the hematocrit and due to the antioxidant activity of the Araticum, it does not seem appropriate to mention that it can have beneficial effects in reversing anemia. I think you should have evaluated more hematological parameters, such as hemoglobin concentrations and red blood cell count, between others.

Lines 204 to 211, “the increase in hematocrit is due to the antioxidant activity” this intrepretation is a quite excessive. In my opinion, you should evaluate more parameters to assess the antioxidant capacity, such as antioxidant enzymes, and relate these parameters.

Lines 223 to 225, it also does not seem factual to mention that the decrease of the number of leukocytes may be associated to anti-inflammatory properties. For that, you should perform other analyzes to assess the anti-inflammatory activity, such as the inhibition of pro-inflammatory cytokines, intercellular adhesion molecules and enzymes (e.g., iNOS, COX-2, 5-LOX), or transcription factors (e.g. NF-κB).

Lines 296 and 300: GPT and TGP should be used only an acronym, because it is the same.

At lines 307 and 308 it is assumed that 3 rats died. In this sense, the total number indicated in the methodology of 60 animals, should it be 57? You should mention the monitorization of  mortality at the methodology part and at the beginning of the results discussion.

Line 320: Is it the reference range for plasma urea values in humans? Please, clarify.

The conclusion should be improved according to the discussion of the results.

Lines 337 and 341. These sentences are not properly supported by the evaluated parameters, as mentioned above. Other analysis should have been carried out to assess anti-inflammatory activity. The decrease of glucose, besides not being significant, presents values within the reference range (~ 70-110 mg / dl) for humans. These conclusions are very excessive.

In conclusion, you should mention which fruits could have the best and worst health effects.

Reviewer 3 Report

They have obtained enough biochemical data, but I do not see results of how much pulp of the different fruits would be necessary to eat a 70kg adult to obtain the necessary benefits for their health such as functional foods and / or food supplements, in addition to the nutrients that each fruit provides . It would also be necessary to know if they should eat the mixed fruits or just one fruit to reap the health benefits. If with the data obtained they can give advice to the population for their cultivation and production of fruits and reach an adequate sustainability of the region. I understand that it is a preliminary study, and I find it interesting especially for the Amazonian population that does not have access to other common foods in the city and these studies can help the Amazonian population to acquire an adequate diet in their region without having to travel to other places further afield in the more developed cities of the country.